# A Novel ZnO/Co$_3$O$_4$ Nanoparticle for Enhanced Photocatalytic Hydrogen Evolution under Visible Light Irradiation

**Tsung-Mo Tien** * and **Edward L. Chen** *

College of Hydrosphere Science & Coastal Water and Environment Center, National Kaohsiung University of Science and Technology, Kaohsiung City 81157, Taiwan
* Correspondence: tmtien@nkust.edu.tw (T.-M.T.); edwardljchen@gmail.com (E.L.C.)

**Abstract:** In recent years, ZnO/Co$_3$O$_4$ nanoparticles (NPs) have been reflected as typical of the most promising photocatalysts utilized in the field of photocatalysis for potentially solving energy shortages and environmental remediation. In this work, a novel ZnO/Co$_3$O$_4$ NP photocatalyst was fabricated and utilized for photocatalytic hydrogen evolution with visible light activity. ZnO/Co$_3$O$_4$ NPs display an improved photocatalytic hydrogen production rate of 3963 μmol/g through a five-hour test under visible light activity. This is much better than their single components. Hence, bare ZnO NPs loaded with 20 wt% Co$_3$O$_4$ NPs present optimum efficiency of hydrogen evolution (793.2 μmol/g/h) with 10 vol% triethanolamine (TEOA), which is 11.8 times that of pristine ZnO NPs. An achievable mechanism for improved photocatalysis is endowed in terms of the composite that promotes the operative separation rate of charge carriers that are produced by visible light irradiation. This study yields a potential process for the future, proposing economical, high-function nanocomposites for hydrogen evolution with visible light activity.

**Keywords:** hydrogen evolution; photocatalytic nanocomposite; spatial charge separation; photoinduced charge carriers; ZnO/Co$_3$O$_4$





## 1. Introduction

Important environmental issues and energy shortage problems have mostly limited the world economy's evolution [1,2]. It is pressing to search for novel technologies for environmental remediation and progress in developing alternative green energy sources. The additive service of fossil fuels, owing to industrial circulation and population increase, has become a critical cause of the harmful pollution issue from global warming and gas disposal [3]. The exploration of a substitute energy source in the future signifies a certain term for durable refinement. In addition, non-green energy properties will decrease over the next few decades. Clear hydrogen (H$_2$) illustrates the clean and solar-light activity of green energy and can be developed from water (H$_2$O). Therefore, the photocatalytic process can be adapted to generate H$_2$ and remove organic compounds from wastewater. In recent years, the photodegradation of environmental pollutants with simultaneous H$_2$ evolution has drawn extensive attention [4–6]. This process could not only generate H$_2$ to aid energy-critical points but also degrade environmental pollutants in wastewater to suppress environmental issues [7–9]. However, the present photocatalytic process still has faults, including the high recombination efficiency of photoinduced charge carriers for single components and the poor solar light application ability for broad bandgap energy photocatalysts [10]. Photocatalytic solar H$_2$ production is one of the most attractive low-cost energy harvesting processes, producing H$_2$ from the photocatalytic activity of water [11]. Thus, it is required to improve the existing photocatalytic routes to boost photocatalytic efficiency.

Numerous semiconductors have been investigated for use as photocatalysts, including metal oxides such as ZnO, TiO$_2$, and Co$_3$O$_4$ [12–14]. However, a single metal oxide photocatalyst displays a rather poor photocatalytic capability in general owing to its confined

solar light absorption and urgent issues of charge carrier recombination caused by poor electron and hole separation capability [15]. To address these matters, heterostructure photocatalysts composed of two or more materials are being explored, suggesting enhanced $H_2$ production capability. Hybrid photocatalysts have a broad range of visible light absorption because of components' suitable bandgap overlap and improved electron and hole separation efficiency between the junction interfaces [16]. In the process, the separation rate of photoinduced charge carriers and the great operation of solar energy are essential factors to enhance the photocatalytic activity. Thus, a great amount of effort has been made to solve these issues. Among different methods, the introduction of heterostructures is a great opportunity for the performance of photoinduced carrier separation in hybrid photocatalysts [17]. The band bending at the heterostructure drives photoinduced charge carrier separation and migration in the opposite direction, restraining electron and hole pair recombination and improving photocatalytic efficiency [18]. Substantially, $Co_3O_4$ has the matched bandgap energy with ZnO to develop a useful step-scheme (S-scheme) heterojunction due to the valence band (VB) potential of $Co_3O_4$ being close to the CB potential of ZnO according to previous studies [19,20]. The formation of S-scheme heterojunctions can contribute to the maximum application of photoexcited electron–hole pairs [21]. On the other hand, to our best knowledge, no study has been conducted on the construction of $ZnO/Co_3O_4$ nano-heterojunctions derived to promote photocatalytic hydrogen evolution efficiency. Furthermore, the formation of merged S-scheme nano-heterojunctions based on $ZnO/Co_3O_4$ is still under challenge.

Currently, the material of nanohybrids is a popular study point owing to its large specific surface area and nanoscale structures that seem ideal for redox reactions [22,23]. In addition, regular semiconductors are mediums that are not well-suited to the essential requirements of the photocatalyst. Zinc oxide (ZnO) and cobalt oxide ($Co_3O_4$) are both transition metal oxides, and ZnO is among the irreducible metal oxides [24]. Particularly, ZnO has a stable hexagonal wurtzite structure with a wide bandgap (3.2 eV) at room temperature. Based on these superior properties, ZnO NPs can be used for various promising purposes [25]. Traditional semiconductors are mainly used as powerful photocatalysts for decomposing harmful pollutants present in textile effluents, owing to their cost-effectiveness, flexible oxidation status, variable bandgap, wider absorption range, and enhanced stability [26]. $Co_3O_4$, as a potential transition metal oxide, is included in several applications, such as hydrogen production, the removal of pollutants, and solar energy conversion [27]. However, pristine $Co_3O_4$ is not a better material for photocatalysts due to its high electron and hole recombination rate, which causes a lower electron and hole pair separation ability and poor photocatalytic performance [28]. However, the wide bandgap of a single component and the quick recombination rate of the photoinduced charge carriers are major barriers to their real utilization in environmental remediation under visible light activity [29]. Different from most traditional semiconductor photocatalysts, ZnO and $Co_3O_4$ nanomaterials have unique electronic and optical properties and a relatively narrow bandgap, which has a good prospect for photocatalytic $H_2$ evolution activity. To enhance the $H_2$ evolution efficiency, sacrificial agents that act as electron donors are adopted to respond to the photoinduced holes [30]. The combination of ZnO and $Co_3O_4$ NPs holds great promise for designing high-efficiency heterojunction photocatalysts due to the fact that both semiconductors are environmentally friendly, chemically stable, and abundant in nature [31]. In this work, the photocatalytic $H_2$ evolution capability of the $ZnO/Co_3O_4$ heterojunction was investigated to research the effect of the $ZnO/Co_3O_4$ interface on the separation rate of photoexcited charge carriers and the photocatalytic $H_2$ evolution activity.

Herein, we report the formation of $ZnO/Co_3O_4$ NPs through a simple hydrothermal treatment. With a broad range of visible light absorption and a quick photoinduced charge carrier mobility rate, the heterojunction exhibited excellent photocatalytic efficiency compared with other ZnO-based photocatalysts. When $Co_3O_4$ was loaded onto ZnO, the advantages of charge accumulation and transfer through the alloy interface showed a lower onset potential and charge transfer resistance, which significantly improved the

photocatalytic activity compared with the use of ZnO and $Co_3O_4$ as separate cocatalysts. Among the various adding ratios of $Co_3O_4$, ZnO/$Co_3O_4$-20 NPs exhibited better efficiency for photocatalytic $H_2$ production, with the total amount of $H_2$ approaching 3965 μmol/g in 5 h. In addition, we further suggested a charge carrier migration route for this heterojunction based on the energy levels from inclusive descriptions. ZnO/$Co_3O_4$ nanohybrids have excellent local photon capture capabilities due to their quantum confinement and surface effects, which are considered excellent light absorbers. This work offers a new perspective on how ZnO/$Co_3O_4$ nanohybrids improve the separation efficiency of photogenerated electron–hole pairs and their application in photocatalytic hydrogen production. It also provides an effective reference for increasing the active sites of heterojunctions.

## 2. Results

### 2.1. Microstructure Characterization

Figure 1 displays the XRD analysis of pristine ZnO, $Co_3O_4$, and ZCo -X NPs. It could be found that each diffraction peak width was narrow and sharp, suggesting great crystallinity. For the pristine ZnO, there were characteristic peaks at 2θ of 31.85°, 34.46°, 36.39°, 47.54°, 56.64°, 62.89°, 66.53°, 68.01°, and 69.26°, corresponding to the characteristics of the (100), (002), (101), (102), (110), (103), (200), (112), and (201) planes of ZnO, respectively (JCPDS 89–1397) [32]. According to the Scherrer equation, the average crystallite size of ZnO was near 8 μm. For $Co_3O_4$, the peaks at the locations of 19.12°, 31.29°, 36.96°, 38.67°, 44.81°, 55.62°, 59.37°, and 65.28° corresponded to the major peaks of the (111), (200), (311), (222), (400), (422), (511), and (440) planes of $Co_3O_4$, respectively (JCPDS 42-1467). Compared to pristine ZnO, a new diffraction peak of 19.1° and 59.3° appeared during the hydrothermal route, which was attributed to the (111) and (511) crystal phases of $Co_3O_4$ [33], verifying the partial transition of ZnO to $Co_3O_4$. For ZCo-X NPs, there were diffraction peaks of both $Co_3O_4$ and ZnO, implying that ZCo-X NPs were well fabricated (Figure 1). Specifically, compared with the XRD pattern of the $Co_3O_4$ sample, the diffraction peak intensities of the (111) and (511) planes of ZnO/$Co_3O_4$ hybrids were enhanced with $Co_3O_4$ addition (see Figure 1 inset).

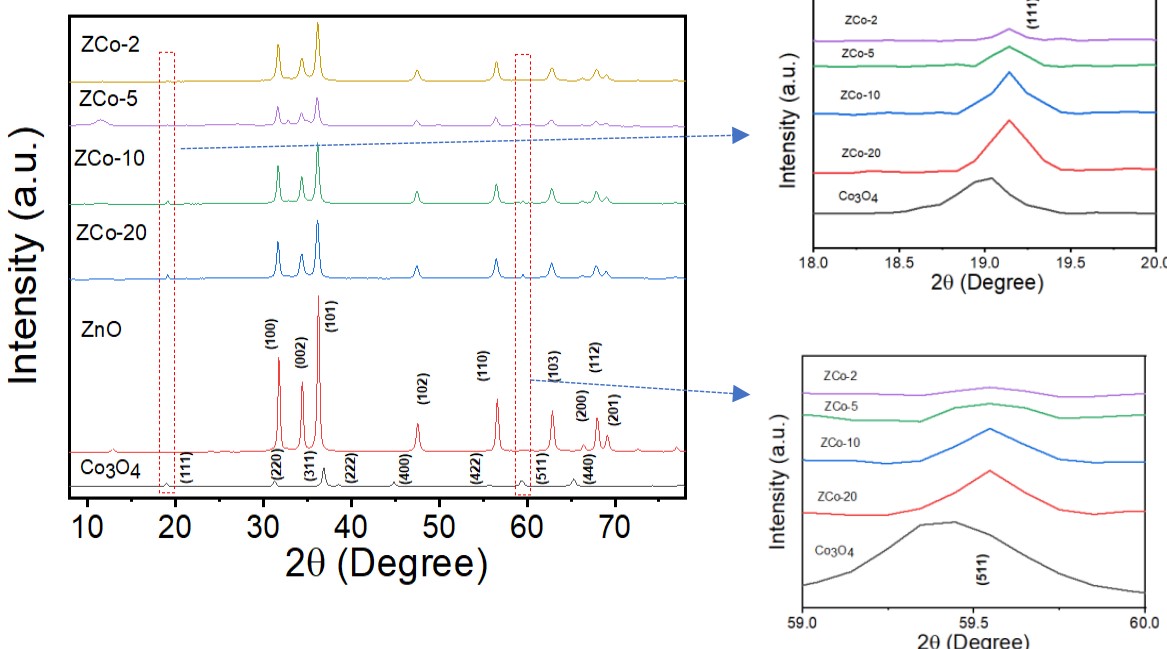

**Figure 1.** XRD patterns of the as-prepared samples.

The morphologies of the as-prepared pristine ZnO, $Co_3O_4$, and ZCo-x NPs are exhibited in Figure 2a–c. As displayed in Figure 2a, sufficient quantities of smooth ZnO with an average particle size of nearly $200 \pm 50$ nm were successfully fabricated. As can be seen in Figure 2b, pristine $Co_3O_4$ displays a sphere-like structure with an average particle size of nearly $350 \pm 50$ nm. Figure 2c presents ZCo-20 NPs composed of various nanoparticles, which are wrinkled with an average particle size of $20 \pm 5$ nm. As exhibited in Figure 2c, there are clusters of $Co_3O_4$ NPs developing on the surface of ZnO. Furthermore, owing to the nanoscale of $Co_3O_4$, there is aggregation of $Co_3O_4$ in the ZCo-x NPs. The ZCo-20 NPs were further confirmed by transmission electron microscopy (TEM) analysis. Figure 2d displays a ZCo-20 NP composed of numerous nanoparticles, which are hemispherical-fashioned particulates of $15 \pm 5$ nm diameter. The HRTEM test of ZCo-20 NPs in Figure 2d displays a well-defined crystallinity phase, and the d-spacing of the $Co_3O_4$ measures 0.35 nm for (111), while the lattice spacing of the (101) plane checked 0.27 nm for the ZnO hexagonal crystal phase (Figure 2e). As the TEM and HRTEM images of the ZCo-20 sample displayed show, the crystallized $Co_3O_4$ NPs have involved the surface of ZnO. Namely, from the HRTEM test in Figure 2e, intimate connections between the $Co_3O_4$ NPs and the ZnO NPs were prepared through the dispersion and hydrothermal routes. According to previous studies concerning ZnO and $Co_3O_4$ heterojunctions [34], the coincidence of heterostructures among ZnO and $Co_3O_4$ might be beneficial to the joining construction, improving the development of $Co_3O_4$ NPs on the surface of ZnO. Due to the electron-tunneling effect, smaller $Co_3O_4$ photocatalysts could promote greater charge carrier transfer capability across the interface than $Co_3O_4$ NPs deposited on the surface of ZnO NPs [35]. The EDS spectrum indicates the conjunction of Co, Zn, and O and the elements' content of ZCo-X NPs. Furthermore, it can be observed from Figure 2f–i that Zn, Co, and O are uniformly distributed. The co-existence of Zn, Co, and O also establishes the configuration of ZCo NPs. The EDS mapping tests verify the even distribution of Zn, Co, and O in ZCo-20 NPs. Especially the Co element displays a lower signal compared to other elements because of the smaller amount added. Furthermore, the line-scanned EDS analysis also checks that $Co_3O_4$ NPs are distributed within the surface of ZnO.

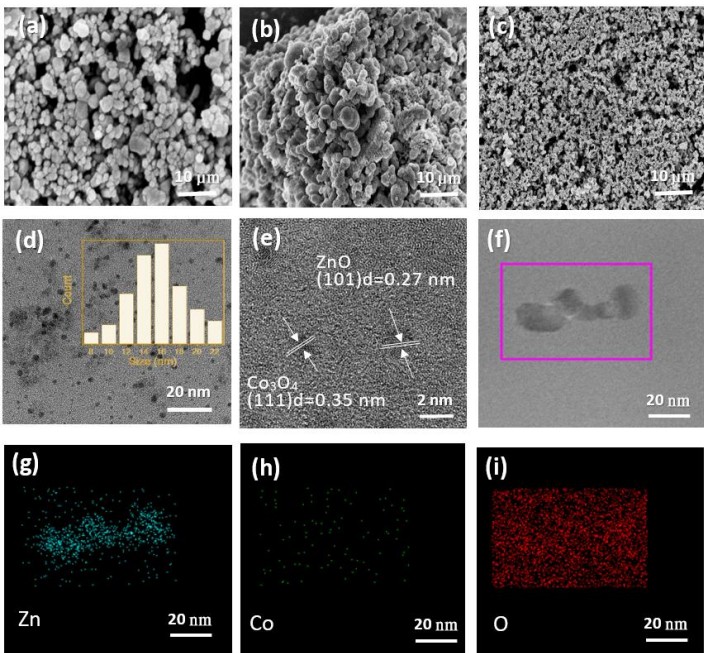

**Figure 2.** SEM images of (**a**) ZnO and (**b**) $Co_3O_4$; TEM images of (**c**,**d**) ZnO/$Co_3O_4$-20; HRTEM tests (**e**); and element mapping images (**f**–**i**) of ZnO/$Co_3O_4$-20.

XPS was used to examine the surface element composition and chemical status. Figure 3a shows the survey spectrum of the ZCo-20 sample. The approach spectra signal of ZCo-20 indicated analogous surface status and composition. A high-resolution Zn 2p spectrum is exhibited in Figure 3b. The core level of the Zn 2p spectrum in Figure 3b depicts two peaks at 1021.56 eV (1021.68 eV) and 1045.59 eV (1044.76 eV) with a spin orbit separation of 24.03 eV (23.08 eV), corresponding to Zn $2p_{3/2}$ and Zn $2p_{1/2}$ of $Zn^{2+}$ in ZnO ($ZnO/Co_3O_4$) [36], respectively. The different binding energies of Zn-O and Co-O result in $ZnO/Co_3O_4$ having a higher energy compared to ZnO and $Co_3O_4$. Particularly, both ZnO and $Co_3O_4$ displayed corresponding binding energies of O 1 s with the absent Zn signals. We assumed that $Co_3O_4$ was entirely embedded by ZnO in the heterostructure. In addition, Figure 3c displays the high-resolution spectrum of Co 2p. The five divided peaks were assigned to Co 2p3/2 and 2p1/2. Compared to pristine $Co_3O_4$, the binding energy of Co in ZCo-20 was altered to a higher energy location, which was congruous with the high-resolution O 1s spectrum in Figure 3d. The altered binding energy was effective proof of the interfacial electric field as the ground-state electron migrated from ZnO to $Co_3O_4$. Consequently, following the above results and analyses, well-fabricated ZCo NPs with tailored morphology and oxygen vacancies were successfully synthesized.

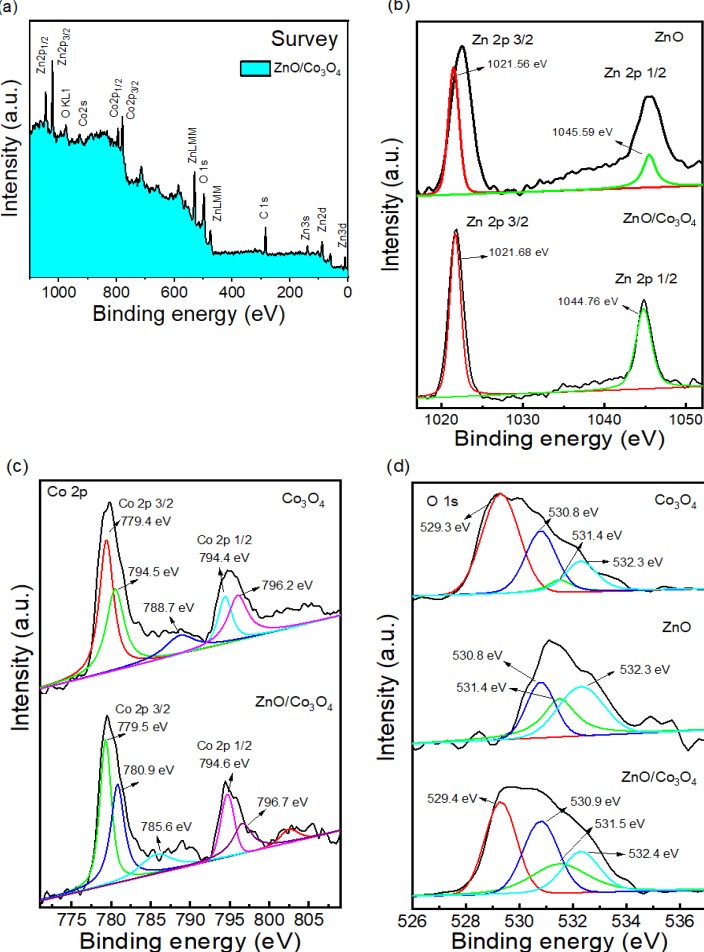

**Figure 3.** XPS of patterns of ZnO, $Co_3O_4$, and ZCo-20. (**a**) Survey XPS spectra; (**b–d**) high-resolution XPS spectra of Zn2p, Co2p, and O 1s.

## 2.2. Photoelectrochemical Behavior

As considered in the photocatalytic $H_2$ production test, the formed morphology of nano-ZnO/Co$_3$O$_4$-20 strongly influences the efficiency of heterostructures. Thus, the visible light utilization and charge carrier separation performance of the compared pristine ZnO and Co$_3$O$_4$ samples were also analyzed. The UV–vis absorption spectrum of ZCo-X was first measured; as exhibited in Figure 4a, ZCo-20 NPs display an obvious improvement in the absorption range from 400 nm to 750 nm correlative to ZCo-2, which is assigned to the quantum confinement effect of the reduced diameter [37]. ZCo-20 exhibits an improved absorbance ability in the visible light region and a slight redshift compared to the ZCo-1 sample. All these changes can lead to differences in the bandgap energy structure and interfacial charge carrier migration of heterostructures. In addition, the variety of ZCo NPs can also influence the visible light utilization of heterostructures, and the absorbance of ZCo-1 NPs is lower than that of ZCo-20 NPs. Furthermore, Figure 4b–d display that ZCo-1 NPs present smaller photocurrent response values, higher EIS Nyquist arc radii, and greater PL intensities than the ZCo-20 NP photocatalyst. As exhibited in Figure 4b, ZCo-1 NPs displayed the highest PL intensity, indicating the photo-response charge carriers recombined quickly. The PL peak of ZCo-20 NPs was much lower than that of ZCo-x NPs, suggesting the recombination of the electrons and holes was dramatically hindered in ZnO/Co$_3$O$_4$-20 NPs [38]. Furthermore, the EIS results showed that ZCo-20 NPs had the smallest arc radius, suggesting that the charge carrier transport resistance of ZCo-20 NPs was the lowest among the four tested photocatalysts (Figure 4c) [39]. The above data and results demonstrated that the quick transfer and separation of the photoinduced charge carriers in the ZCo-20 NPs exhibited higher photocurrent density compared with ZCo-20 NPs, implying enhanced separation performance of the induced charge carriers in ZCo-1 NPs (Figure 4d). According to Figure 4a, ZCo-20 NPs displayed better visible light adsorption efficiency. However, ZCo-1 NPs had a lower photocurrent density (Figure 4d), which could be attributed to the recombination rate of photoinduced charge carriers increasing beyond the confined bandgap value. In addition, the results suggest that the nanoscale of Co$_3$O$_4$-doped and especially O$_2$ vacancies in ZCo-20 NPs play major roles in enhancing the photoinduced electron and hole pair separation ability and photocatalytic $H_2$ production efficiency of ZCo-20 NPs.

Electron paramagnetic resonance (EPR) was performed using 5,5-dimethyl-1-pyrrolin-N-oxide (DMPO) as a trapping agent to compare free radicals developed on the surface of the ZCo-20 NP photocatalyst during the photocatalytic activity route. The results are shown in Figure 5. There are no signals found under the dark condition for the photocatalyst, suggesting visible light activity should be essential for the generation of free radicals. During the illumination of visible light, the twofold feature peaks of DMPO-$\cdot$O$_2^-$ signals emerge in the sample, and the intensity of the characteristic peaks is improved with the extended irradiation time. It is a key factor that the peak intensity of $\cdot$O$_2^-$ for ZCo-20 NPs under light is obviously better than the sample in the dark condition, which demonstrates that a greater amount of $\cdot$O$_2^-$ free radicals form on the surface of ZCo-20 NPs. The results indicate the sample can generate $\cdot$O$_2^-$ under visible light activity, and ZCo-20 NPs have better photocatalytic reduction efficiency.

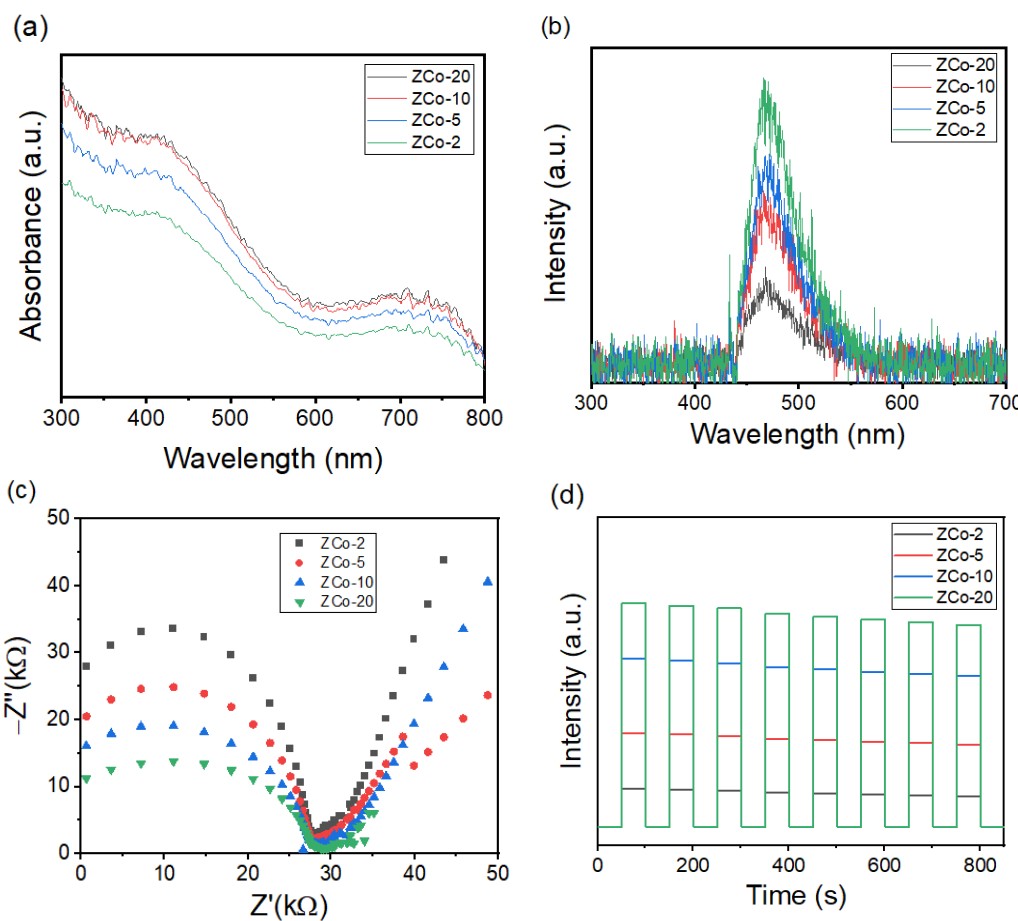

**Figure 4.** (**a**) UV–vis spectra, (**b**) PL spectra, (**c**) EIS Nyquist plots, and (**d**) transient photocurrent responses of ZCo-2, ZCo-5, ZCo-10, and ZCo-20 samples.

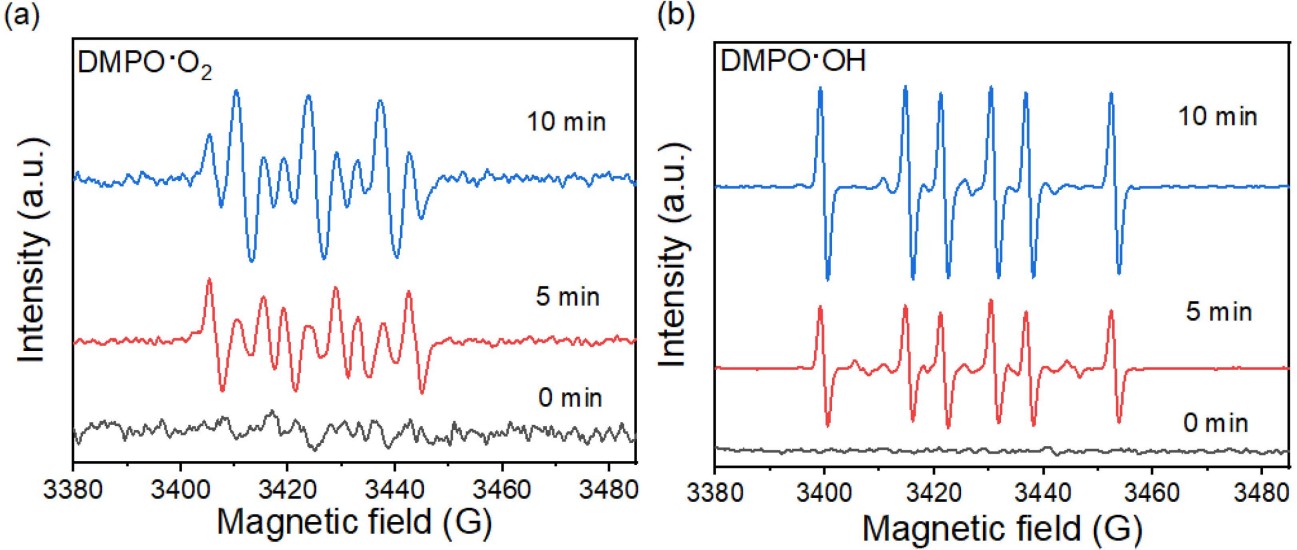

**Figure 5.** EPR spectra of radical adducts DMPO-$\cdot O_2^-$ (**a**) and DMPO-$\cdot$OH (**b**) in ZCo-20 NPs under dark and visible light activity at different times at room temperature.

### 2.3. Hydrogen Production Behavior

Due to the fact that the composition of different ratios in heterostructures has a significant effect on the photocatalytic system, the ratio of $Co_3O_4$ to ZCo was optimized, and the connected results can be observed in Figure 6. As noted, the optimized mass ratio of $Co_3O_4$ to ZCo could be checked to be 20 wt%. Additionally, in order to reflect the excellence of ZCo-x NPs, photocatalytic $H_2$ production-optimized ZCo-x NPs were compared (Figure 6a). As displayed in Figure 6a, ZnO and $Co_3O_4$ express poor $H_2$ production of nearly 336.5 and 539.1 $\mu mol \cdot g^{-1}$ within 5 h, respectively. In addition, ZCo-x NPs present an obvious improvement in $H_2$ production (2352.3, 3448.2, 3757.2, and 3965.5 $\mu mol \cdot g^{-1}$ within 5 h). Under the same time conditions, ZCo-x NPs generate a better $H_2$ evolution of 3965.5 $\mu mol \cdot g^{-1}$, which may be caused by the improved migration and separation performance of photoinduced electrons and holes in ZCo-x NPs. The photocatalytic $H_2$ production also verifies the elevated $H_2$ production efficiency of ZCo-x NPs (Figure 6b). As exhibited, the $H_2$ production efficiency of ZCo-20 NPs is 793.2 $\mu mol \cdot g^{-1} \cdot h^{-1}$, which is near 11.8 and 7.4 times higher than that of pristine ZnO and $Co_3O_4$ samples, respectively. The better photocatalytic efficiency of ZCo-20 NPs further exceeds most of the studied ZnO-based heterostructure photocatalysts. Then, the number of ZCo-20 NPs added was further optimized (Figure 6c). As the dosage of the sample increased, the entire surface area became larger. Accordingly, the number of photocatalysts further increased, which then improved the absorption of visible light for the production of charge carriers and enhanced the efficiency of the generation of reactive oxygen species (ROS), superoxide, and hydroxyl radicals for better hydrogen production [40]. This behavior may be attributed to the increasing sample dosage. Visible light penetration decreases owing to the turbidity caused, which refers to the accumulation of photocatalysts [41]. As exhibited in Figure 6a, during the same conditions of the photocatalytic system, hydrogen production efficiency approaches its best capability when the dose of the sample is 20 mg, which may be due to the photon absorption of 20 mg of sample that can approach its maximum. When screening the curves for all photocatalysts of $ZnO/Co_3O_4$, it is significant to suggest that the concentration of hydrogen produced does not increase linearly but starts to rise faster only after the first or second hour. This implies that the photocatalyst is activated after a positive time of response or contact with the sacrificial reagents. As it is known, from the aspect of energy conversion, the highlight of the photocatalytic activity is tied together with the number of absorbed photons [42]. A favorable photocatalyst procedure is useful for photon absorption; too few samples conduct the weak operation of visible light, and too many samples conduct a shading effect in the photocatalytic activity [43]. Generally speaking, according to the connection between $H_2$ production efficiency and the mass of photocatalysts, ZCo-20 NPs demonstrate the highest $H_2$ evolution efficiency of 793.2 $\mu mol \cdot g^{-1} h^{-1}$ (at 20 mg of catalyst), which is nearly 2.5 and 1.7 times higher than that of 2 mg (312.6 $\mu mol \cdot g^{-1} h^{-1}$) and 5 mg (456.8 $\mu mol \cdot g^{-1} h^{-1}$) of sample, respectively. This indicates that the greatest performance is 1.7 times greater than that of ZCo-2 NPs, implying that $Co_3O_4$ plays a significant role in improving the photocatalytic $H_2$ production efficiency. To further explore the practical utilization value, cycling tests of the optimized ZCo-20 NPs were performed. The photocatalytic stability of ZCo-20 NPs was further measured and is shown in Figure 6d. As depicted, there is a slight decline in $H_2$ production efficiency after four continuous cycles in a total of 20 h (about 11% decrease), suggesting the profitable stability of ZCo NPs and a better potential for effective utilization. Figure 6e displays the XRD pattern of ZCo-20 NPs before use and after four cycling tests. It can be clearly seen that the intensities of the feature peaks of the photocatalyst only have negligible decreases, and the locations of these feature peaks are practically the same before and after use. As depicted, ZCo-20 NPs after photocatalytic activity display a phase structure and morphology that are almost identical to those of ZCo-20 NPs before the photocatalytic reaction, indicating the good structural and compositional stability of the ZCo-20 NP photocatalyst.

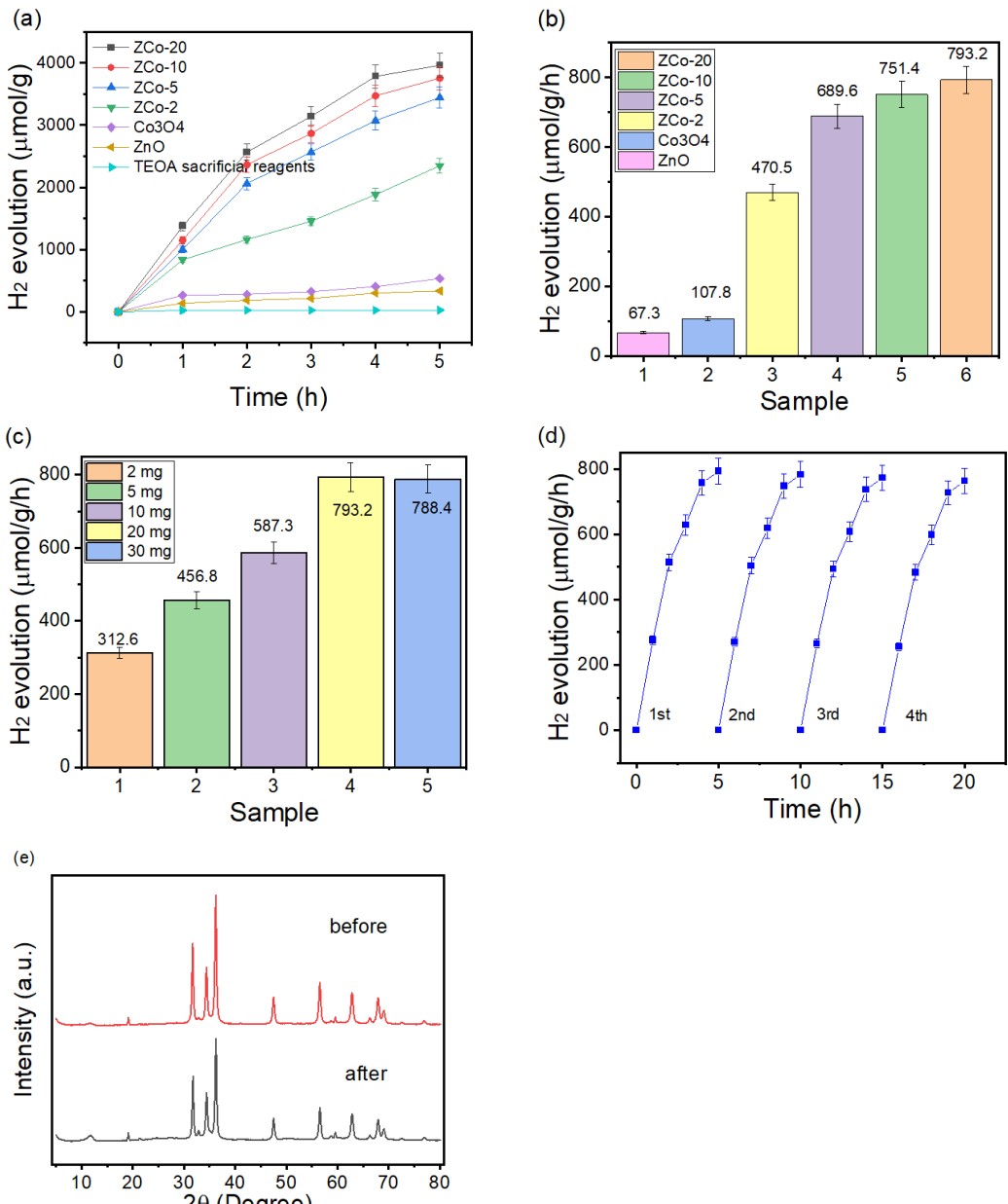

**Figure 6.** (**a**) $H_2$ production amount at different irradiation times; (**b**) $H_2$ production efficiency of various photocatalysts; (**c**) $H_2$ production efficiency with various amounts of ZCo-20; (**d**) $H_2$ production recycling stability test; (**e**) XRD analysis (before and after use).

## 3. Materials and Methods

### 3.1. Preparation of Materials

All reagents were purchased from Merck and Sigma Aldrich (A.R grade, St. Louis, MO, USA) with no subsequent decontamination. Zinc acetate dihydrate ($Zn(CH_3COO)_2 \cdot 2H_2O$, >98%) and cobalt acetate ($Co(CH_3COO)_2 \cdot 4H_2O$, >97%) were used as $Zn^{2+}$ and $Co^{2+}$ precursors, respectively. Ethanol absolute ($CH_3CH_2OH$, >99%), hydrochloric acid (HCl, 37%), glacial acetic acid ($CH_3COOH$, >99.7%), isopropyl alcohol (IPA), p-benzoquinone (BQ), ethylenediaminetetraacetic acid (EDTA), and deionized water (D.I.) were used throughout this work. All chemicals were used as obtained without any further purification.

### 3.2. Synthesis

The $Co_3O_4$ NPs were fabricated using $Co(CH_3COO)_2 \cdot 4H_2O$ by hydrothermal process. A total of 0.4 M of $Co(CH_3COO)_2 \cdot 4H_2O$ was added to 100 mL of ethylene glycol and mixed with 50 mL of D.I. water. The above solutions were mixed using a magnetic stirrer for 50 min, and 0.3 M sodium carbonate solution was dissolved and then stirred for 40 min. Then, the provided solutions were heated at 190 °C for 4 h in a Teflon-lined autoclave. Subsequently, the received mixture solution was washed with ethanol and D.I. water several times, then dried at 100 °C for 8 h. The received product was calcinated at 500 °C for 3 h for further use. Afterward, 0.3 M of $Zn(CH_3COO)_2 \cdot 2H_2O$ and 100 mL of absolute alcohol were mixed by continuously stirring for 50 min. Next, 0.1 M of NaOH was dissolved, and the mixture suspension was delivered to a Teflon-lined stainless steel autoclave heated at 150 °C for 5 h. Furthermore, the mixture solution was cooled to room temperature and then washed with ethanol and D.I. water several times. The resulting sample was dried at 70 °C for 12 h and further calcinated at 500 °C for 4 h. Finally, the ZnO NPs were acquired after cooling.

$ZnO/Co_3O_4$ NPs were fabricated by hydrothermal processes. First, ZnO NPs (5 mmol) and $Co_3O_4$ NPs (5 mmol) were dissolved into 100 mL of absolute alcohol and stirred for 40 min. The obtained mixture was taken into an autoclave with a volume of 50 mL to carry out the hydrothermal process at 170 °C for 20 h. The final product was separated from the autoclave once it was cooled to room temperature. Finally, the product was washed with ethanol and D.I. water alternatively three times. The received product was dried in a vacuum oven at 80 °C for 12 h, and thus the $ZnO/Co_3O_4$ (ZCo) NPs were synthesized. To prepare the ZCo NPs as a series of ZCo-x NPs (x = 2%, 5%, 10%, and 20%), different amounts of $Co_3O_4$ were added to the initial mixture solution, following a similar recipe for ZnO.

### 3.3. Characterization

The crystal structure and chemical composition were investigated by X-ray diffraction (XRD) on the Bruker D8 Advanced (Billerica, MA, USA). The chemical information and bond state of every element were determined by X-ray photoelectron spectroscopy (XPS, Thermo Fisher Scientific, Waltham, MA, USA). Electron paramagnetic resonance (EPR) was used on the Bruker A300 (Munich, Germany) at 77 K to check active radicals under visible light, where 5,5-Dimethyl-l-pyrroline N-oxide (DMPO) was used as the trapping agent to trap the hydroxyl radical ($\cdot$OH) or the superoxide radical ($\cdot O_2{}^-$). The microstructures and morphologies of the products were recorded using a scanning electron microscope (SEM, ZEISS AURIGA, Oberkochen, Germany) and a transmission electron microscope (TEM, JEOL JEM-2100F, Tokyo, Japan). Ultraviolet–visible diffuse reflectance (UV–vis) spectrum in the range of 200–800 nm was characterized using a spectrophotometer (Hitachi UV-4100, Tokyo, Japan). Fluorescence spectroscopy was performed at room temperature with the F-7000 Edinburgh Analytical Instruments fluorescence spectrometer (Tokyo, Japan) to evaluate the carrier separation qualitatively.

### 3.4. Photocatalytic Hydrogen Evolution Tests

The $H_2$ produced in the closed system was quantitatively measured by GC 7890 gas chromatography (Agilent, Santa Clara, CA, USA) with a TCD detector. A 20 mg sample was entirely dispersed in 50 mL aqueous solution including 10 vol% TEOA by ultrasound for 20 min, and then moved to a reactor. The reaction solution was maintained at 10 °C with cooling water and magnetic stirring, and the reaction process was vacuumed before visible light activity. Afterward, a 350 W Xe lamp with simulated solar illumination AM 1.5G (100 mW/$cm^2$) and an ultraviolet cut-off filter ($\lambda > 420$ nm) were used to irradiate the reactor. Prior to the tests, the mixture solutions were magnetically stirred for 20 min. At set intervals, 3 mL of the solution was sampled. Then, the clear supernatant was obtained by centrifugation. Stability tests were determined with the same conditions, but the closed

reactor needed to be degassed to decrease the amount of $H_2$ produced before starting each cycle.

*3.5. Photoelectrochemical Tests*

The photoelectrochemical analysis was recorded on an electrochemical workstation with a standard three-electrode service. The working electrodes were arranged as follows: 2 mg of sample was dispersed in a mixed solution including 10 μL of Nafion, 50 μL of ethanol, and 500 μL of water, and a 300 W Xe lamp was used as the light source. Following an ultrasound process for 20 min, the resultant suspension of 100 μL was added and dried on a FTO substrate ($1 \times 1$ cm$^2$). The FTO glass was used as the working electrode, and the saturated Pt electrode and $Hg/Hg_2Cl_2$ electrode were carried out as the counter electrode and reference electrode, respectively.

## 4. Conclusions

In this work, a well-designed $ZnO/Co_3O_4$ NP photocatalyst with intimate interface contact was fabricated for the first time, and the resultant NP heterojunction indicated a significant improvement and stability in photocatalytic $H_2$ production efficiency. Thus, the $H_2$ production efficiency of the $ZnO/Co_3O_4$ NP heterojunction was significantly enhanced to 793.2 μmol·g$^{-1}$ h$^{-1}$, which was 11.8 and 7.4 times higher than that of pristine ZnO and the $Co_3O_4$ samples, respectively. This study has demonstrated the heterojunction interfacial contact of $ZnO/Co_3O_4$ NPs, which could serve as a new route for the interface engineering of the photocatalytic heterojunction process. The synergistic effects between ZnO and $Co_3O_4$ enhance the absorption ability of visible light irradiation and the separation performance of photo-response electron and hole pairs, indicating the excellent redox efficiency of the photocatalysts. The great photocatalytic efficiency of the NP heterojunction for $H_2$ production efficiency can be attributed to the extended absorption range, the improved interaction between ZnO and $Co_3O_4$, the enhanced migration and separation efficiencies of photoinduced electron and hole pairs, the increased number of sites with photocatalytic activity, and the powerful redox reaction.

**Author Contributions:** Writing—original draft, Methodology, T.-M.T.; Writing—review and editing, E.L.C.; Data curation, Formal analysis, T.-M.T.; Funding acquisition, T.-M.T.; Writing—review and editing, T.-M.T.; Validation, E.L.C.; Resources, T.-M.T.; Formal analysis, Data collection, E.L.C.; Conceptualization, Writing—review and editing, E.L.C.; Supervision, Investigation, E.L.C.; Supervision, Project administration, T.-M.T. All authors have read and agreed to the published version of the manuscript.

**Funding:** This research received no external funding.

**Data Availability Statement:** Not available.

**Acknowledgments:** The MOST and the NKUST are gratefully acknowledged for their general support. The authors gratefully acknowledge the use of HRTEM equipment belonging to the Instrument Center of National Cheng Kung University.

**Conflicts of Interest:** The authors declare no conflict of interest.

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
