# Peer review of "A Novel ZnO/Co3O4 Nanoparticle for Enhanced Photocatalytic Hydrogen Evolution under Visible Light Irradiation"

_catalysts, doi:10.3390/catal13050852_

Round 1

Reviewer 1 Report

The publication concerns "A novel ZnO/Co3O4 nanoparticles for enhanced photocatalytic hydrogen evolution under visible light irradiation". The concept, and the methodology are not original. Several inaccuracies and inconsistencies in the text must be corrected. For example, in the abstract H2 evolution is given at 793.2 µmol/h but in the manuscript it is 793.2 µmol/g/h. Consistency of units is the basis of science. Line 6, the band gap of ZnO is not 3.37 eV but 3.2eV. The state of the art is partially presented in the introduction. The objectives and the concept are not well described. Preparation materials, synthesis characterization paragraphs are pretty well described. For the photocatalytic test, the spectrum of the lamp must be presented when one claims to do photocatalysis in the visible. Just mentioning the use of a filter is not enough. In the results  paragraph, a serious shortcoming is the bad decomposition treatment of the XPS spectra which does not allow to draw relevant information. Finally, the other shortcomings in the results of photocatalysis are numerous. fig 6a shows the evolution of H2 which is not linear which is commonly the proof of a stoichiometric reaction and not photocatalyst. There is no material balance between the hydrogen produced and the amount of TEOA used. Performance data is reported by mass of catalyst. However, there is no study of the activity-mass influence. However, it is known that the activity increases almost linearly up to a plateau. Without this information the comparison of catalysts is meaningless. The conclusions are more phenomenological than explanatory.

Author Response

Author's Reply to the Review Report (Reviewer 1)

The publication concerns "A novel ZnO/Co3O4 nanoparticles for enhanced photocatalytic hydrogen evolution under visible light irradiation". The concept, and the methodology are not original. Several inaccuracies and inconsistencies in the text must be corrected. For example, in the abstract H2 evolution is given at 793.2 µmol/h but in the manuscript it is 793.2 µmol/g/h. Consistency of units is the basis of science.

Reply: We appreciate the comments from this reviewer. We have revised and added as follows above. It has been modified in Abstract: “ photocatalytic hydrogen production rate of 3963 μmol/g through a five-hour test under visible light activity. This is much great than their single components. Hence, bare ZnO NPs loaded with 20 wt% Co3O4 NPs present optimum efficiency of hydrogen evolution (793.2 μmol//g/h) with 10 vol% triethanolamine (TEOA),”

Line 6, the band gap of ZnO is not 3.37 eV but 3.2eV.

Reply: We appreciate the comments from this reviewer. We have been revised and added as follow above. It has been described in Page 2, Lines 69.

The state of the art is partially presented in the introduction. The objectives and the concept are not well described.

Reply: It’s true that our original statement is not clearly enough. According to your suggestion, we have added some additional description and cited some references in the revised manuscript. It has been described in the introduction (red color) of the revised manuscript.

Preparation materials, synthesis characterization paragraphs are pretty well described. For the photocatalytic test, the spectrum of the lamp must be presented when one claims to do photocatalysis in the visible. Just mentioning the use of a filter is not enough.

Reply: Many thanks for giving us so many suggestions. It has been described in Page 3, Lines 150~151:  “a 350 W Xe lamp with simulated solar illumination AM 1.5G (100 mW/cm2) and an ultraviolet cut-off filter (λ > 420 nm) was performed to irradiate the reactor.”

In the results paragraph, a serious shortcoming is the bad decomposition treatment of the XPS spectra which does not allow to draw relevant information.

Reply: We appreciate the comments from this reviewer. We have modified Fig. 3 in the revised manuscript.

Fig. 3. XPS of patterns of ZnO, Co3O4 and ZCo-20. (a) Survey XPS spectra, (b-d) The high-resolution XPS spectra of Zn2p, Co2p and O 1s.

Finally, the other shortcomings in the results of photocatalysis are numerous. fig 6a shows the evolution of H2 which is not linear which is commonly the proof of a stoichiometric reaction and not photocatalyst. There is no material balance between the hydrogen produced and the amount of TEOA used. Performance data is reported by mass of catalyst. However, there is no study of the activity-mass influence. However, it is known that the activity increases almost linearly up to a plateau. Without this information the comparison of catalysts is meaningless. The conclusions are more phenomenological than explanatory.

Reply: Thanks for your advised. It has been described in Page 8-9, Lines 370~386:  “ As the dosage of the sample increased, the entire surface area became larger. Accordingly, the number of photocatalysts further increased, which then improved the absorption of visible light for the production of charge carriers and enhanced the efficiency of generation of reactive oxygen species (ROS), superoxide and hydroxyl radicals for better hydrogen production [40]. This behavior may be assigned to the increasing sample dosage. The visible light penetration reduces owing to the turbidity caused, which refers to the accumulations of photocatalysts [41]. As exhibited in Fig. 6a, during the same conditions of photocatalytic system, the hydrogen production efficiency approaches the best capability when the dose of the sample was 20 mg, which may be owing to the photon absorption of 20 mg of sample that can approach the maximum. When screening the curves for all photocatalysts of the ZnO/Co3O4, it is significance to suggest that the concentration of made hydrogen does not enlarge linearly, but only after the first or second hour it raise faster. This implies that the photocatalyst is activated after a positive time of response or contact with the sacrificial reagents. As we know, from the aspect of energy conversion, the highlight of the photocatalytic activity is tied together with the number of absorbed photons [42]. A favorable photocatalyst procedure is useful for photon absorption, too slight samples conduct the weak operation of visible light, and too much sample conducts a shading effect in the photocatalytic activity [43].”

Fig. 6. (a) H2 production amount at different irradiation time.

  • Altaa, S. H. A.; Alshamsi, H. A. H.; Al-Hayder, L. S. J. Synthesis and characterization of rGO/Co3O4 composite as nanoadsorbent for Rhodamine 6G dye removal, Water Treat. 2018, 114, 320.
  • Shathy, R.A.; Fahim, S. A.; Sarker, M.; Quddus, M. S.; Moniruzzaman, M.; Masum, S. M.; Molla, M. A. I. Natural sunlight driven photocatalytic removal of toxic textile dyes in water using B-doped ZnO/TiO2 nanocomposites, Catalysts 2022, 12 (3), 308.
  • Chen, J.; Chuang, Y. J. Directly electrospinning growth of single crystal Cu2ZnSnS4 nanowires film for high performance thin film solar cell, J. Power Sources, 2013, 241, 259.
  • Qureshi, M.; Takanabe, K. Insights on Measuring and Reporting Heterogeneous Photocatalysis: Efficiency Definitions and Setup Examples. Mater. 2017, 29, 158.

Reviewer 2 Report

The authors reported a ZnO/Co3O4 NPs for photocatalytic hydrogen production. The characterizations are abundant, and the composites should a better activity. However, the analysis should be improved. This manuscript can be accepted after solving the following questions.

1.        Two hydrogen production rates are wrong in the abstract part.

2.        With scale of 10 μm in Figure 2c, it is hard to obtain the conclusion of particle size of 20 nm, please using a high resolution one with smaller scale, such as 20 nm. From Figure 2c, it is hard to observe the Co3O4 on the surfaces of ZnO. Mapping data should be redone.

3.        Several important references related to hydrogen production should be referred (Renewable and Sustainable Energy Reviews, 2020, 132, 110040; Molecules, 2022, 27, 4241; Chem. Eng. J., 2023, 456, 141032).

4.        The XPS should be reanalyzed. For example, wrong conclusion of “ZnO illustrated five peaks with the binding energy of 1021.5 eV, 1022.4 eV, 1023.5 eV, 1044.6 eV and 1045.6 eV, corresponding to 2p3/2 and 2p1/2 232 of Zn2+ respectively.” was obtained.

5.        Format problems, such as “b-d)” should be changed to “(b-d)” in Line 261.

6.        Wrong caption for Figure 5. The corresponding description should be rewrite.

7.        How to prove that the existence of heterostructure?

Author Response

Author's Reply to the Review Report (Reviewer 2)

The authors reported a ZnO/Co3O4 NPs for photocatalytic hydrogen production. The characterizations are abundant, and the composites should a better activity. However, the analysis should be improved. This manuscript can be accepted after solving the following questions.

Two hydrogen production rates are wrong in the abstract part.

Reply: Thanks for your advised. We have revised in the Abstract. “ photocatalytic hydrogen production rate of 3963 μmol/g through a five-hour test under visible light activity. This is much great than their single components. Hence, bare ZnO NPs loaded with 20 wt% Co3O4 NPs present optimum efficiency of hydrogen evolution (793.2 μmol//g/h) with 10 vol% triethanolamine (TEOA), ”

With scale of 10 μm in Figure 2c, it is hard to obtain the conclusion of particle size of 20 nm, please using a high resolution one with smaller scale, such as 20 nm. From Figure 2c, it is hard to observe the Co3O4 on the surfaces of ZnO. Mapping data should be redone.

Reply: Thanks for your suggestions. The Figure 2 has been revised as follow above.

Fig. 2. SEM images of (a) ZnO, (b)Co3O4, TEM images of (c-d) ZnO/Co3O4-20; HRTEM tests, (e) and element mapping images (f-i) of ZnO/Co3O4-20.

Several important references related to hydrogen production should be referred (Renewable and Sustainable Energy Reviews, 2020, 132, 110040; Molecules, 2022, 27, 4241; Chem. Eng. J., 2023, 456, 141032).

Reply: Sincerely thank the reviewer for your comment. We carefully read the references and made an in-depth literature review about hydrogen production. The references have been added (ref), and changes have been incorporated into the revised manuscript.

Ref.

  • Zhao, W.; Chen, Z.; Yang, X.; Qian, X.; Liu, C.; Zhou, D.; Sun, T.; Zhang, M.; Wei, G.; Dissanayake, P. D.; Ok, Y. S. Recent advances in photocatalytic hydrogen evolution with high-performance catalysts without precious metals, Sustain. Energy Rev. 2020, 132, 110040.
  • Hu, N.; Cai, Y.; Li, L.; Wang, X.; Gao, J. Amino-Functionalized Titanium Based Metal-Organic Framework for Photocatalytic Hydrogen Production, Molecules 2022, 27, 4241.
  • Zhang, Y.; Wang, X.; Ren, X.; Luo, S.; Huang, H.; Chen, R.; Shao, S.; Liu, D.; Gao, J.; Gui, J.; Ye, J. Building rapid charge transfer channel via engineering Ni coordinated flexible polymer for efficient solar hydrogen evolution, Eng. J. 2023, 456, 141032.

The XPS should be reanalyzed. For example, wrong conclusion of “ZnO illustrated five peaks with the binding energy of 1021.5 eV, 1022.4 eV, 1023.5 eV, 1044.6 eV and 1045.6 eV, corresponding to 2p3/2 and 2p1/2 of Zn2+ respectively.” was obtained.

Reply: Thank you very much for pointing out this. It’s true that our original statement is not clearly enough. According to your suggestion, we have revised the Figures (XPS) in the revised manuscript. It has been described in Page 5, Lines 243~245: “The core level of the Zn 2p spectrum in Fig. 3b depicts two peaks at 1021.56 eV (1021.68 eV) and 1045.59 eV (1044.76 eV) with a spin orbit separation of 24.03 eV (23.08 eV), corresponding to Zn 2p3/2 and Zn 2p1/2 of Zn2+ in ZnO (ZnO/Co3O4)[X39], respectively.”

Fig. 3. (c) The high-resolution XPS spectra of Zn2p.

  • Ahmad, I.; Shukrullah, S.; Naz, M. Y.; Bhatt, H. N.; Ahmad, M.; Ahmed, E.; Ullah, S.; Hussien, M. Recent progress in rare earth oxides and carbonaceous materials modified ZnO heterogeneous photocatalysts for environmental and energy applications, Environ. Chem. Eng. 2022, 10, 107762.

Format problems, such as “b-d)” should be changed to “(b-d)” in Line 261.

Reply: Thank you for your valuable suggestion. We have modified as following above in the revised manuscript.

Wrong caption for Figure 5. The corresponding description should be rewrite.

Reply: Thank you very much for pointing out this matter. We have been revised and modified in the figure caption: “Fig. 5. EPR spectrum of radical adducts DMPO-O2‾ (a) and DMPO-OH (b) in ZCo-20 NPs under dark and visible light activity with different time at room temperature. ”

How to prove that the existence of heterostructure?

Reply: Thank you very much for pointing out this. Constructing heterojunctions with other semiconductor materials is a strategy to effectively improve the separation performance of carriers and maintain light absorption [S1, S2]. The construction of heterojunction can not only enhance the stability of photocatalyst, but also improve the light capture performance [S3, S4]. Recently, nano-heterostructure have found attractive applications in the fields of photoelectrochemistry. As a kind of heterojunction photocatalyst, direct Z-scheme photocatalyst has aroused great interest because of its effective charge separation and redox ability [S5]. Unlike the conventional photocatalyst, the conduction band minimum (CBM) and the valence band maximum (VBM) of the direct Z-scheme photocatalyst don’t have to cross the redox potential of water [S6]. This means that it has a narrower band gap, which expands the collection range of sunlight while satisfying a high redox capacity. Under visible light, photogenerated electrons with stronger catalytic reduction ability are retained in the conduction band (CB) of one side of the semiconductor, while photogenerated holes with stronger catalytic oxidation ability are retained in the valence band (VB) of the other side of the semiconductor, which increases its redox ability to decompose water [S7].

In this study, ZnO/Co3O4 nano-heterojunction photocatalyst was synthesized by a simple hydrothermal method. The nano-heterostructure was verified by SEM (Fig.2) and TEM (Fig.2), and its elemental composition was analyzed by XRD (Fig.1), XPS (Fig.3) and Mapping tests (Fig.2).

[S1] Y. Wu, Y. Li, L. Zhang, Z. Jin, NiAl-LDH in-Situ Derived Ni2P and ZnCdS Nanoparticles Ingeniously Constructed S-Scheme Heterojunction for Photocatalytic Hydrogen Evolution, ChemCatChem 2022, 14.

[S2] J. Li, M. Li, Y. Li, X. Guo, Z. Jin, Lotus-Leaf-Like Bi2O2CO3 Nanosheet Combined with Mo2S3 for Higher Photocatalytic Hydrogen Evolution, Sep. Purif. Technol. 2022, 288.

[S3] Z. Fan, X. Guo, Z. Jin, X. Li, Y. Li, Bridging Effect of S-C Bond for Boosting Electron Transfer over Cubic Hollow CoS/g-C3N4 Heterojunction toward Photocatalytic Hydrogen Production, Langmuir: ACS J. Surfaces Colloids 2022, 38, 3244–3256.

[S4] K. Wang, H. Xie, Y. Li, G. Wang, Z. Jin, Anchoring Highly-Dispersed ZnCdS Nanoparticles on Nico Prussian Blue Analogue-Derived Cubic-Like NiCoP Forms an S-Scheme Heterojunction for Improved Hydrogen Evolution, J Colloid Interface Sci 2022, 628, 64–78.

[S5] Q. Xu, L. Zhang, J. Yu, S. Wageh, A.A. Al-Ghamdi, M. Jaroniec, Direct Z-scheme photocatalysts: principles, synthesis, and applications, Mater. Today 2018, 21, 1042–1063.

[S6] A. Singh, M. Jain, S. Bhattacharya, MoS2 and Janus (MoSSe) based 2D van der Waals heterostructures: emerging direct Z-scheme photocatalysts, Nanoscale Adv. 2021, 3, 2837–2845.

[S7] X. Liu, W. Kang, L. Qi, J. Zhao, Y. Wang, L. Wang, W. Wang, L. Fang, M. Zhou, Two-dimensional g-C3N4/Ti2CO2 heterostructure as a direct Z-scheme photocatalyst for water splitting: a hybrid density functional theory investigation, Physica E Low Dimens 2021, 134.

Reviewer 3 Report

Authors demonstrate photocatalytic activity of ZnO/Co3O4 nanoparticles for hydrogen evolution under visible light irradiation.

Results presented are classical for the investigation of photocatalysts using a wide range of methods for their characterization and proper investigations of photocatalytic activity and stability.

Unfortunately, there is not comparison with another photocatalysis. It will be better if authors demonstrate advantages not only with pristine oxides ZnO and Co3O4.

The statement that optimized mass ratio of Co3O4 to ZnO is 20wt% looks strange because there were not experiments with concentrations more than 20%.

In any case article could be published with some improvement.

Some comments.

11.      Abstract. It is not clear “hydrogen production rate of 3963 μmol/g” in what time. It should be mentioned from what water solutions the H2 evolution was investigated.

22.      Introduction. Line 72. It is not clear the phrase “ZnO and Co3O4 nanomaterials have been explored for the photocatalytic H2 evolution activity, band gaps energy of the photocatalysts should be through the reduction promising of H2O”.

33.      Line 139 . Phrase “Photocatalytic activity of H2 production was determined in a gas chromatography (GC-7890)” is not correct . Moreover, Lines 145 and 146 repeat this and are situated in more proper place of the description of photocatalytic experiments.

44.      Check following lines

- Line 136   (punctuation mark ?).

- Line 121 ( pristine ZnO/Co3O4 ?).

- Line 173 (What partial transition of ZnO to Co3O4 authors took in mind?).

Author Response

Author's Reply to the Review Report (Reviewer 3)

Authors demonstrate photocatalytic activity of ZnO/Co3O4 nanoparticles for hydrogen evolution under visible light irradiation. Results presented are classical for the investigation of photocatalysts using a wide range of methods for their characterization and proper investigations of photocatalytic activity and stability. Unfortunately, there is not comparison with another photocatalysis. It will be better if authors demonstrate advantages not only with pristine oxides ZnO and Co3O4.

The statement that optimized mass ratio of Co3O4 to ZnO is 20wt% looks strange because there were not experiments with concentrations more than 20%. In any case article could be published with some improvement. Some comments.

Abstract. It is not clear “hydrogen production rate of 3963 μmol/g” in what time. It should be mentioned from what water solutions the H2 evolution was investigated.

Reply: Thanks for your advised. It has been described in the Abstract: “ photocatalytic hydrogen production rate of 3963 μmol/g through a five-hour test under visible light activity. This is much great than their single components. Hence, bare ZnO NPs loaded with 20 wt% Co3O4 NPs present optimum efficiency of hydrogen evolution (793.2 μmol/h) with 10 vol% triethanolamine (TEOA), ”

Introduction. Line 72. It is not clear the phrase “ZnO and Co3O4 nanomaterials have been explored for the photocatalytic H2 evolution activity, band gaps energy of the photocatalysts should be through the reduction promising of H2O”.

Reply: Thank you very much for pointing out this. According to your suggestion, we have added some additional description in the revised manuscript. It has been described in Page 2, Lines 79~81: “Different from most traditional semiconductor photocatalysts, ZnO and Co3O4 nanomaterials have unique electronic and optical properties, and have a relatively narrow band gap, which has a good prospect in photocatalytic H2 evolution activity.”

Line 139 . Phrase “Photocatalytic activity of H2 production was determined in a gas chromatography (GC-7890)” is not correct . Moreover, Lines 145 and 146 repeat this and are situated in more proper place of the description of photocatalytic experiments.

Reply: Thank you very much for pointing out this. According to your suggestion, we have added some additional description in the revised manuscript. It has been described in Page 3, Lines 145~146: “The H2 produced in the closed system was quantitatively measured by GC 7890 gas chromatography with a TCD detector.”

Check following lines

- Line 136   (punctuation mark ?).

- Line 111 ( pristine ZnO/Co3O4 ?).

- Line 173 (What partial transition of ZnO to Co3O4 authors took in mind?).

Reply: Thank you very much for pointing out this. We have revised it as follows above. It was modified on Page 3, Line 153, and Page 3, Line 129 within the revised manuscript.

In addition, the measured diffraction peaks are quite consistent with the ZnO standard card (JCPDS 89–1397) and Co3O4 standard card (JCPDS 42-1467). It is noteworthy that the diffraction pattern from bulk Co3O4 sample contains intense peaks from (111) and (511) planes and other peaks with lesser intensity from (200), (311), (222), (400), (422), and (400) planes, thus suggesting the diffraction from Co3O4. On the other hand, the XRD patterns of the ZnO samples indicate the intense peak from (100), (002), (101) planes and other peaks with lesser intensity from (102), (110), (103), (200), (112), and (201) planes, thus suggesting the diffraction from ZnO samples. Besides, the feature peaks (111) and (511) planes for all ZnO/Co3O4 photocatalysts moved to lower 2-theta angles compared with the standard JCPDS cards. This particular preliminarily demonstrated that the ZnO in Co3O4 nanohybrids underwent a tensile deformation in the crystalline phase of nano-heterojunctions. According to your suggestion, we have added some additional descriptions in the revised manuscript. It has been described in Page 4, Lines 178~180: “ Specifically, compared with the XRD pattern of Co3O4 sample, the diffraction peak intensities of (111) and (511) planes of ZnO/Co3O4 hybrids were enhanced with  Co3O4 adding (see Fig. 1 inset).”

Figure 1. XRD patterns of the as-prepared samples

Round 2

Reviewer 1 Report

Responses to comments are satisfactory. There are still a few errors to be corrected in the text. The manuscript can be accepted with minor corrections.

Author Response

Reply: Special thanks to you for your good comments. I have checked the revised manuscript (red color) and refined the language and format carefully. 

Reviewer 2 Report

 (793.2 μmol//g/h) should be changed to (793.2 μmol/g/h) in the abstract.

Author Response

Reply: Thanks for the kind suggestion. We have revised it to 793.2 μmol/g/h and show it in the revised manuscript.